# A Systematic Review of Treatments of Post-Concussion Symptoms

**DOI:** 10.3390/jcm11206224

**Published:** 2022-10-21

**Authors:** Camille Heslot, Philippe Azouvi, Valérie Perdrieau, Aurélie Granger, Clémence Lefèvre-Dognin, Mélanie Cogné

**Affiliations:** 1AP-HP, GH Paris Saclay, Hôpital Raymond Poincaré, Service de Médecine Physique et de Réadaptation 104, Boulevard Raymond Poincaré, 92380 Garches, France; 2CESP, Inserm, Paris-Saclay University, UVSQ, 94807 Villejuif, France; 3Rehabilitation Unit, Rennes University Hospital, 2 Rue Henri Le Guilloux, 35000 Rennes, France

**Keywords:** mTBI, post-concussion, rehabilitation, cognition, psychological care, physical

## Abstract

Approximately 10–20% of patients who have sustained a mild Traumatic Brain Injury (mTBI) show persistent post-concussion symptoms (PCS). This review aims to summarize the level of evidence concerning interventions for PCS. Following the PRISMA guidelines, we conducted a systematic review regarding interventions for PCS post-mTBI until August 2021 using the Medline, Cochrane, and Embase databases. Inclusion criteria were the following: (1) intervention focusing on PCS after mTBI, (2) presence of a control group, and (3) adult patients (≥18 y.o). Quality assessment was determined using the Incog recommendation level, and the risk of bias was assessed using the revised Cochrane risk-of-bias tool. We first selected 104 full-text articles. Finally, 55 studies were retained, including 35 that obtained the highest level of evidence. The risk of bias was high in 22 out of 55 studies. Cognitive training, psycho-education, cognitive behavioral therapy, and graded return to physical activity demonstrated some effectiveness on persistent PCS. However, there is limited evidence of the beneficial effect of Methylphenidate. Oculomotor rehabilitation, light therapy, and headache management using repetitive transcranial magnetic stimulation seem effective regarding somatic complaints and sleep disorders. The preventive effect of early (<3 months) interventions remains up for debate. Despite its limitations, the results of the present review should encourage clinicians to propose a tailored treatment to patients according to the type and severity of PCS and could encourage further research with larger groups.

## 1. Introduction

Approximately 42 million people worldwide and 150,000 in France are affected by an mTBI yearly [1,2,3]. According to the WHO Collaborating Task Force [4], the three main current criteria used to define mTBI are a Glasgow Coma Scale (GCS) score between 13 and 15 (30 min after the injury), a Post-Traumatic Amnesia (PTA) duration of less than 24 h, and/or a loss of consciousness lasting less than 30 min. After mTBI, 10–20% of the population will retain aftereffects called post-concussion Symptoms (PCS). According to the International Statistical Classification of Diseases and Related Health Problems (ICD-10), PCS is defined as a “Head injury usually sufficiently severe to result in loss of consciousness and then development within four weeks of at least three of the eight following symptoms: headache, dizziness, fatigue, irritability, sleep problems, concentration problems, memory disorders and emotion perturbations” [5]. Individuals who have sustained an mTBI have significantly more self-reported cognitive symptoms than controls up to 4 years after injury [6,7,8,9,10]. PCS is associated with anxiety and post-traumatic stress disorder (PTSD) in 16 to 26.8% of cases 6 months after mTBI [11,12,13] but other factors, such as vestibular/cervical dysfunction, may also contribute to outcomes [14]. Poor outcomes are difficult to predict at an individual level [10] but have been found to be statistically predicted by a combination of variables, including pre-injury factors (older age, lower education, pre-existing mental health problems), injury-related factors (assault-related injury, lower GCS score, presence of intracranial lesion), and early post-traumatic psychological factors (emotional distress, maladaptive coping) [6,15].

Several guidelines on the management of concussion/mTBI and PCS are available, created by expert groups, such as the Ontario Neurotrauma Foundation (https://braininjuryguidelines.org/concussion/ accessed on 15 October 2022) or the Centers for Diseases Control and Prevention (CDC; https://www.cdc.gov/traumaticbraininjury/mtbi_guideline.html accessed on 15 October 2022). Expert recommendations include early psycho-education, telephone counseling, graded return to physical activity, psychological treatment, cognitive rehabilitation, and vestibular or oculomotor rehabilitation [16]. However, despite the large number of publications in the field, the efficacy of interventions for PCS is still uncertain and remains a matter of debate (for a recent review, see ref. [16]). Moreover, PCS include a wide constellation of cognitive, psychological and somatic complaints, and it is difficult, based on the work published to date, to state which type of intervention is effective for which type of symptoms. A systematic review and meta-analysis on non-pharmacological interventions for persistent PCS was published recently and concluded that “based on very low to low certainty (…) the guideline panel found weak scientific support for commonly applied non-pharmacological interventions to treat persistent PCS” [16]. However, this latter review selected only 19 randomized controlled trials and did not include pharmacological interventions nor interventions related to specific symptoms associated with PCS, such as headache, fatigue, or associated mood disturbances.

The present review aims at summarizing the available evidence on the treatment of PCS after mTBI, including pharmacological interventions, and according to the type of symptom targeted (cognitive, physical, psychological). To this end, the proposed systematic review aimed at answering the following questions: Are pharmacological and non-pharmacological interventions, including early interventions, efficient in reducing PCS after mTBI? What are the risks of bias in the existing studies? Additionally, what is the level of quality of the evidence?

The review questions were defined using the PICO methodology: Population: adult participants with PCS after mTBI/concussion; Intervention: early intervention and all pharmacological and non-pharmacological interventions; Comparison: usual care, wait-list group, or any other intervention; Outcome: cognitive, physical, and psychological measures.

## 2. Materials and Methods

Study eligibility criteria. A systematic literature review was conducted in accordance with the PRISMA guidelines (www.prisma-statement.org accessed on 15 October 2022). We selected articles published in English until August 2021 and including treatment of adult individuals with PCS after mTBI. Previous literature reviews were not included, but studies quoted within these reviews were also included if relevant. Criteria for inclusion were the following: (1) intervention studies focusing on PCS after concussion/mTBI, (2) studies including a control group, and (3) studies focusing on adult patients (18 years or more). Studies including both mTBI and moderate to severe TBI were excluded (except if data regarding mTBI could be analysed separately). Studies including adolescents were included only if adolescent participants represented a minority of the study sample. For this review, we included only studies including a control group, but we chose to present studies matching our other inclusion criteria except for the control group in the Appendix A in order to contribute to our discussion. Both pharmacological and non-pharmacological interventions were included, and there was no specific time frame for intervention.

Sources. The Medline, Cochrane, and Embase databases were searched. The keywords used were: «mild traumatic brain injury» AND «rehabilitation», «treatment», «post-concussion», «post commotional», OR «post-concussive». Then, we added the following keywords one by one: «attention», «memory», «dysexecutive», «executive», «neglect», «social cognition», «anxiety», «mood», «phobia», «post-traumatic stress disorder», «headache», «migraine», «irritability», «concentration», «insomnia», «sleep», «fatigue», «dizziness», «balance», OR «vestibular syndrome». Articles were then independently selected on the basis of title and abstract screening by two authors (CH and MC). If there was a disagreement between these two authors, a third author (PA) intervened to reach a consensus. The following flow chart (Figure 1) illustrates the article selection procedure.

Data extraction and quality assessment. Studies were divided into four groups according to the main target or timing of intervention: cognitive symptoms and PCS in general, mood and sleep disorders, somatic complaints and fatigue, and early interventions (within three months since injury). Information extracted (Table A1, Table A2, Table A3 and Table A4) included: experimental design, age and gender of the participants, etiology of mTBI, time since injury, the main objective of the study (including nature of the intervention and of control treatment when applicable), the number of participants, outcomes and tools, and the main results. Quality assessment was based on the Incog grading system: [17] A = Recommendation supported by at least one meta-analysis, systematic review, or randomized controlled trial of appropriate size with a relevant control group; B = Recommendation supported by cohort studies that at minimum have a comparison group, well-designed single-subject experimental designs, or small sample size randomized controlled trials; C = Recommendation supported primarily by expert opinion based on their experience, or uncontrolled case series without comparison groups. The risk of bias in the included studies was assessed by two authors (MC and PA) using the Cochrane Rob2 tool revised version [18]. This tool considers five domains that can be a source of bias: randomization process, deviations from intended interventions, outcome data, measurement of the outcome, and selection of the reported results. Each domain is assessed as having a low, high, or unclear risk of bias. Upon judgment of the risk of bias for each domain, the authors judged the overall risk of bias for each publication included.

Studies selected (Figure 1). First, 3896 abstracts were identified using the keywords «mild traumatic brain injury» (in the title or abstract) AND «rehabilitation», «treatment», «post-concussion», «post commotional», OR «post-concussive» (in all fields). Regarding cognitive symptoms after mTBI, 1399 records were identified by adding the following keywords one by one: “attention”, “memory”, “dysexecutive”, “executive”, “neglect”, “concentration”, OR “social cognition”. Lastly, 18 studies were included. Regarding psychological symptoms, 1626 records were identified by adding the following keywords one by one: “anxiety”, “mood”, “affect”, “phobia”, “phobic”, “insomnia”, “sleep”, “irritability”, OR “post-traumatic stress disorder”. In the end, four studies were included and all of them concerned sleep disorders. Regarding physical symptoms after mTBI, 871 records were identified by adding the following keywords one by one: “fatigue”, “dizziness”, “balance”, “vertigo”, “headache”, «migraine», OR “vestibular syndrome”. Finally, 15 studies were included. Regarding early interventions after mTBI, we selected from 18 articles from previous research related to intervention in a delay of 3 months or less after mTBI.

Overall, 104 articles matched our inclusion criteria and were selected. Eight of these papers were excluded as participants were not restricted to patients with mTBI (some of them included moderate or severe TBI or patients with PTSD without TBI). The main manuscript will only focus on papers rated as A or B level of evidence (55 articles). The remaining 41 articles (uncontrolled group studies or isolated case reports) are presented as Appendix A. The main characteristics of the 55 selected studies are shown in Table A1, Table A2, Table A3 and Table A4, with their corresponding level of evidence.

## 3. Results

### 3.1. Treatments Focusing on Cognitive Symptoms and the Reduction of PCS in General

Overall, we selected 18 Grade A or B studies that focused on cognitive complaints after mTBI and/or reduction of PCS in general, based on different methods. Different methods such as cognitive training, psycho-education, pharmaceutical treatments, non-invasive brain stimulation, hyperbaric oxygen, technology-assisted rehabilitation, and others, were proposed.

#### 3.1.1. Cognitive Training Programs and Psycho-Education

Five Grade A or B randomized controlled trials (RCT) based on cognitive rehabilitation and/or psycho-education were selected. All of these trials found positive results. One of them, found in ref. [19], included 89 participants in an interdisciplinary 22-week program (S-REHAB) with exercise therapy and physiotherapeutic coaching, showing a reduction of PCS measured with the Rivermead Post-Concussion Symptoms Questionnaire (RPQ) immediately post-treatment and 6 months later. A 10-week program utilizing group-based compensatory cognitive training in 119 veterans also found a significant improvement in attention, learning abilities, and executive functioning [20]. Tiersky et al. [21] combined cognitive remediation and psychotherapy for 20 participants with mTBI and PCS lasting more than one year (including two patients with moderate TBI) and found a significant reduction of anxiety and depression and an improvement in divided attention at 1 and 3 months follow-up. However, there was no significant improvement in community integration scores. Another study [22], including 112 participants, showed that an 8-week interdisciplinary, individually tailored intervention based on a gradual return to activities and principles from cognitive behavior significantly reduced PCS at 3 months. Novakovic-Agopian et al. [23] found some efficacy in a goal-oriented attentional self-regulation training program performed during 5 weeks by 40 veterans with comorbid PTSD and mTBI on cognitive functions, emotional regulation, and functional performance. Nevertheless, another study [24] compared different cognitive rehabilitation interventions (psycho-education, computer-based training, therapist-directed manualized training, and integrated therapist-directed training combined with cognitive-behavioral psychotherapy) but did not find any difference between techniques concerning the primary outcome criterion, the Paced Auditory Serial Addition Test. In conclusion, most of these five RCTs reported significant positive effects on cognitive functions, although there is no evidence that one single program is more effective than the other. In addition, three uncontrolled studies reported positive effects of various cognitive interventions, including Attention Process Training-II or Goal Management Training (see Appendix A).

#### 3.1.2. Pharmacological Treatments

Five Grade A or B publications reported the positive effects of three drug treatments (Guanfacine, Methylphenidate, and Enzogenol) on cognitive symptoms after mTBI, but three of these articles were based on the same trial at different time-points [25,26,27]. 

One of them [28] found an improvement in working memory (evaluated by an “N-back” task) and increased activation of the frontal lobe in functional MRI 2.5 h after the administration of alpha2-antagonists (Guanfacine) in 13 patients. Another [25] found a positive effect of Methylphenidate on fatigue and processing speed (but not on pain) with a dose-dependent effect in 51 patients (including 4 with moderate TBI). They followed 30 participants who had reported positive effects during the initial phase of this latter study and were treated with methylphenidate for a further 6 months and still showed significant improvement, as compared to baseline, on mental fatigue, depression, anxiety, processing speed, attention, and working memory [26]. A follow-up study was then conducted by the same group over a period of approximately 5.5 years in 17 patients [27]. A comparison was made between those who had continued or discontinued Methylphenidate. Treatment was associated with improvement in mental fatigue, depression, anxiety, and processing speed, suggesting that the Methylphenidate effect is maintained over time but reversible if discontinued [27]. Lastly, administration of Enzogenol (pine extract) during 6 weeks significantly reduced self-perceived cognitive impairments (evaluated with the Cognitive Failures Questionnaire) in 62 participants [29].

To summarize, five studies using pharmaceutical treatments reported positive results on the outcome (mainly with Methylphenidate). In addition, one uncontrolled study found a beneficial effect of antidepressant treatment (see Appendix A).

#### 3.1.3. Non-Invasive Brain Stimulation

Only one randomized controlled trial was selected in this category, but five additional uncontrolled studies were reported (see Appendix A). Moussavi et al. [30] found that participants who had an injury < 12 months who received active rTMS (repetitive Transcranial Magnetic Stimulation) showed significant improvements in the RPQ compared to those in the same subgroup who received sham stimulation and to those with a longer duration of injury (>14 months) who received active rTMS.

#### 3.1.4. Hyperbaric Oxygen Therapy

Five Grade A randomized controlled trials evaluating the effect of hyperbaric oxygen (HBO2) treatment on cognitive functions in patients with mTBI found contradictory results (two positive, three negative). Harch et al. [31] found that 63 participants with 150 Kpa HBO2 significantly improved on a wide range of measures, including cognition, mood, PCS, sleep, and quality of life. In another study including 56 patients [32], HBO2 improved cognitive functions and quality of life. However, three other randomized sham-controlled trials [33,34,35] did not find any cognitive improvement after HBO2 in a large sample of participants suffering from PCS after mTBI. 

In conclusion, overall, these studies do not support the use of HBO2 at this time for persistent PCS after mTBI. 

#### 3.1.5. Technology-Assisted Cognitive Rehabilitation

Two studies were selected, one positive and one negative. Belanger et al. [36] did not find any significant effect of a web-based educational intervention regarding TBI (severity, symptoms and their management, expectation of recovery) on symptom severity or occurrence in 158 participants. However, it should be noted that subgroup analyses suggested some benefit in the group of patients receiving concurrent mental health treatment at baseline who reported significantly less PCS than controls at 6 months. Cooper et al. [24] found an improvement in auditory attention assessed using the Paced Auditory Serial Addition Test after several interventions (psycho-education, computer-based Cognitive Réhabilitation (CR), therapist-directed manualized CR, and integrated therapist-directed CR combined with cognitive-behavioral psychotherapy) in 126 participants. 

In addition, four uncontrolled studies were found, assessing the effect of computerized attentional training, a self-run computer-based psycho-education program, and a videophone-based therapy combined with psycho-education or assistive technology aids (such as IPads with scheduling application, iPhone with a sample voice memo, or electronic list), with mixed results (see Appendix A).

#### 3.1.6. Other Rehabilitation Techniques

Five uncontrolled studies were included in this category, including diverse forms of intervention, such as mindfulness-based stress reduction, a targeted treatment on specific symptoms/impairments (such as psychological, sleep, ocular, vestibular symptoms), head-eye vestibular motion therapy, musical training, or Qigong practice (see Appendix A). Although positive findings were reported, the level of evidence within the studies found was very low.

### 3.2. Treatments Focusing on PTSD, Mood, and Sleep Disorders

We selected four studies (three Grade A) that specifically focus on sleep disorders, with mixed results.

In addition, we found six studies specifically addressing PTSD after mTBI with encouraging results, but they were all uncontrolled studies (see Appendix A).

A large (*n* = 356) study using telephone-based problem-solving treatment [37] found a significant improvement in the sleep quality of participants with mTBI at 6 months, but the effect was not maintained 12 months post-injury. Another study [38] showed that cognitive behavioral therapy significantly reduced sleep disturbance with a moderate effect in 24 participants after a mild to moderate TBI. However, there were no significant group differences in objective sleep quality, cognitive functioning, post-concussion symptoms, or quality of life, possibly due to a lack of statistical power. Furthermore, 6-week morning blue-light therapy compared to placebo light in 35 adults with an mTBI under 18 months showed a significant improvement in sleep, daytime sleepiness, depression, PCS, and executive functions [39,40].

### 3.3. Treatment Focusing on Somatic Complaints and Fatigue

We selected 14 studies in this category (six Grade A). 

#### 3.3.1. Balance Disorders

The treatment of balance disorders and vertigo was evaluated in four controlled studies and in nine uncontrolled or case-series studies (see Appendix A). Kleffelgaard et al. [41] reported on the efficacy of 8-week group-based vestibular rehabilitation intervention in 65 participants, which appeared to speed up recovery for patients with dizziness and balance problems. However, the benefits were not maintained 2 months after the end of the intervention. A randomized study of 21 patients suffering from balance deficit after mTBI lasting more than 12 months evaluated the effect of video game therapy on the X-Box 360 console compared to balance platform therapy [42]. It showed that both groups improved in Community Balance and Mobility Scale scores, but only the video game therapy group improved on the Unified Balance Scale and Timed Up and Go test. Schneider et al. [43] found some efficacy in a combination of 8-week vestibular rehabilitation and cervical spine physiotherapy with a decrease in the time until medical clearance to return to sport in individuals with prolonged PCS at the subacute stage. Lastly, a study of 71 patients found some efficacy of 40 sessions over 12 weeks of hyperbaric oxygen therapy on balance impairments [44].

In summary, there is some, although limited, evidence to suggest that specific training may improve dizziness and balance disorders after mTBI.

#### 3.3.2. Headache and Migraine

The management of post-traumatic headaches or migraines after mTBI was evaluated in seven controlled studies and five uncontrolled studies (see Appendix A).

Four controlled studies focused on the effect of rTMS on headaches with mostly positive results. Leung et al. reported two positive studies of rTMS applied on the left motor cortex [45] or on the left dorsolateral prefrontal cortex [46] to alleviate post-traumatic headache (prevalence and pain intensity) 1 and 4 weeks post-treatment. Another study, including 20 participants, used 10 sessions of rTMS therapy applied to the left dorsolateral prefrontal cortex with mixed results, with effects which fell below clinical significance thresholds [47]. Choi et al. [48] found that 10 sessions of rTMS applied to the primary motor cortex of the affected hemisphere significantly reduced pain intensity after treatment and after 1, 2, and 4 weeks. Even if large randomized controlled trials seem necessary to confirm these preliminary results, rTMS seems promising for reducing headaches in mTBI with PCS.

Kjeldgaard et al. [49] found no significant effect of 9 weeks of cognitive behavioral therapy on 90 participants for headache and pain perception in comparison to a waiting list group. Two sessions of manual therapy on the neck were tested in a small RCT (*n* = 23) in comparison to cold packs on the neck. Treatment was associated with a significant reduction of pain index 2 weeks after the end of treatment, but this effect was no more statistically significant 5 weeks later [50]. Esterov et al. (2021) [51] showed a reduction in pain in 26 participants assessed using a visual analogic scale after an osteopathic manipulative treatment, but no significant difference regarding pain was found within the questionnaire results. 

In addition, five uncontrolled studies evaluated the drug treatment of post-concussive headaches with drugs such as botulinum toxin, gabapentin, tricyclics, epidural injection of saline, and oxygen or monoclonal antibodies targeting Calcitonin gene-related peptide, but the level of evidence of these studies remains very low (see Appendix A).

#### 3.3.3. Oculomotor Disorders

The treatment of oculomotor and vision disorders after mTBI was addressed in three controlled studies with mixed results (and in several additional uncontrolled studies, as can be seen in Appendix A). Thiagarajan et al. reported in two separate papers the results of a 6-week cross-over trial comparing oculomotor rehabilitation with placebo training in 12 patients with near-vision symptoms [52,53]. During each session, all three oculomotor subsystems (vergence/accommodation/version) were trained. They found an improvement in vergence and of near work-related symptoms and visual attention [52], and of oculomotor control, reading rate, and overall reading ability [53]. Hyperbaric oxygen therapy did not show any effect on eye-tracking abnormalities in 60 patients with mTBI compared to a sham-control treatment [54]. 

In summary, although some preliminary positive results have been reported, the level of evidence of oculomotor rehabilitation remains low.

#### 3.3.4. Post-Injury Fatigue

We found only one Grade A study. Kolakowsky-Hayner et al. [55] included 123 patients to evaluate the effect of a graduated physical activity program (home-based walking program using a pedometer to track a daily number of steps accompanied by tapered coaching calls over a 12-week period) compared with a control condition (nutritional counseling with the same schedule of coaching calls). The results showed less reported fatigue at the end of the active part of the intervention (24 weeks) and after a wash-out period (36 weeks).

### 3.4. Early Interventions

Eighteen controlled studies assessed the effect of an early intervention (<3 months post-injury) designed to prevent the occurrence and chronicity of PCS. However, there were mixed findings.

Seven publications reported positive results. Mittenberg et al. [56] showed that based on a printed manual and a consultation with a therapist (providing psycho-education, information, techniques for reducing symptoms, and instructions for a gradual return to premorbid activities), significantly shorter symptom duration, fewer symptoms, fewer symptomatic days, and lower severity levels were found in 58 participants after very early intervention (prior to hospital discharge). Ponsford et al. [57] found that in 202 participants, there were positive effects of providing an information booklet outlining the symptoms associated with mTBI and suggestions of coping strategies one week after the injury. Three months post-injury, patients in the intervention group reported significantly fewer symptoms and were less stressed than those in the control group. Bell et al. [58] showed the positive impact of focused, scheduled telephone counseling (five phone calls) on the mean symptom score of 366 participants, reducing the proportion of patients reporting each individual symptom (except anxiety) and issues with daily functioning. Caplain et al. [59] proposed that for 80 participants in the early stage post-injury (<1 month) presenting high risk factors for persistent PCS, multidimensional intervention with 14 sessions combining psycho-education and cognitive rehabilitation was desirable. They found a significantly decreased risk of persistent PCS at the 6-month follow-up. The preventive effect of early CBT (<6 weeks) was found to be effective by Silverberg et al. [60] in 28 patients, decreasing the risk of persistent PCS at 3 months. The effect size on PCS reduction was moderate. The use of a text messaging-based education and behavioral support was associated with a non-significant trend for a decreased report of irritability, anxiety, headaches, and concentration difficulties within 14 days post-trauma (n = 43) [61]. One quasi-experimental non-randomized study found that social work intervention (providing reassurance and education regarding the recovery process and follow-up guidelines, including brief alcohol intervention) at the acute stage post-mTBI at the emergency department significantly reduced alcohol use 3 months post-injury [62].

However, contrasting results were also reported in nine publications. No significant difference was found in self-reported symptom severity between bed rest and no bed rest for 107 patients during the first 10 days after mTBI [63]. Ghaffar et al. [64] included 191 participants within one-week post-injury who were randomly assigned to multidisciplinary treatment or no treatment. They did not find any group difference at 6 months (although, in subjects with a psychiatric history, the provision of treatment was associated with significantly fewer depressive symptoms). Andersson et al. [65] offered an early individualized, tailored multidisciplinary outpatient rehabilitation program involving physiotherapists, occupational therapists, and social workers after mTBI. Patients had repeated outpatient appointments and thereafter, telephone contacts. There was no significant difference between the control group one year post-injury. Heskestad et al. [66] assessed the effect of very early (2-week post-injury) educational intervention based on one single consultation focusing on cognitive counseling, advice, information, and reassuring. They failed to find any significant difference compared to the control group regarding symptoms, depression, sleep, and fatigue at 3- and 6-month follow-ups, but there was a very high dropout rate (85%). An early intervention visit (14 to 21 days after the trauma) in addition to written information and treatment did not show more efficacy than treatment alone on symptom level at 3 months in a study including 97 patients [67]. Vikane et al. [68] did not find any significant between-group differences regarding the 12-month return to work following a multidisciplinary outpatient follow-up program for patients being at-risk or sick-listed with persistent PCS. However, there were fewer post-concussion symptoms in the intervention compared to the control group at 12 months. Varner et al. [69] found no beneficial effect of an intervention within 24 h post-injury based on cognitive rest and graduated return to usual activity discharge instructions (n = 118). Post-commotional symptoms were not significantly different in the intervention group compared to controls 2 and 4 weeks post-injury. Early intervention (cognitive behavioral intervention with psycho-education on mTBI and enhancement of the sense of self-control) compared to telephone counseling in “at risk” patients failed to find any significant difference in return-to-work anxiety and depression [70]. Paradoxically, in this latter study, telephone counseling was associated with fewer complaints and more frequent full recovery at 12-month follow-up than cognitive behavioral intervention. Audrit et al. [71] did not find any significant time x group interaction of a psycho-educative and counseling intervention (SAAM) on PCS assessed using the RPQ.

In addition, two early drug trials may be mentioned here. Early Cerebrolysin (a nootropic drug which has been found useful to improve cognitive function in patients with Alzheimer’s disease) therapy (within 24 h) improved cognitive functions, especially long-term memory and visuo-constructive functions in 32 patients 3 months after mTBI [72]. However, Atorvastatin, which was assumed to improve cerebral plasticity, was administered for 7 days in 52 patients with mTBI at an early stage (within 1 week) but did not show any significant effect on PCS (evaluated by the RPQ) 3 months later [73].

In summary, it is difficult to conclude at this stage as results differ from one study to the other. Prolonged bed rest should not be recommended, but the beneficial effect of an early educational or multidisciplinary intervention remains to be debated.

### 3.5. Risk of Bias Assessment

Among the 55 RCT selected, 22 studies were judged to have an overall high level of risk of bias, 26 studies raised some concerns regarding the risk of bias, and only 7 studies had a low risk of bias (please consider Figure A1, Figure A2, Figure A3 and Figure A4 in Appendix B).

## 4. Discussion

The objective of the present systematic review was to identify therapeutic approaches which may improve persistent PCS after mTBI.

### 4.1. Treatment of Cognitive Symptoms and Reduction of PCS in General

Among the five studies based on cognitive training and/or psycho-education, four reported positive results, which are quite encouraging and constitute a relatively good level of evidence to recommend such training in patients with persistent PCS, including those with associated PTSD. However, the level of evidence of alternative treatments, such as pharmaceutical drugs, non-invasive brain stimulations (in particular rTMS), hyperbaric oxygen, or technology-assisted rehabilitation, remains low or inconsistent, and none of these approaches can be recommended for clinical practice at this stage, even if some promising results have been found with rTMS.

### 4.2. Treatment of PTSD, Mood, and Sleep Disorders

Regarding mood disorders and PTSD, we could not find any controlled study specifically addressing these issues in participants with mTBI, but positive findings were reported in a few uncontrolled studies (see Appendix A) or in samples including, but not limited to, mTBI. For example, Ponsford et al. [74] found some efficacy in a 9-week CBT for reducing anxiety and depression in 75 participants after mild to severe TBI. Four controlled studies focusing on sleep disorders were selected, with encouraging but contrasting results, so the level of evidence remains modest. Cognitive-behavioral therapy and psychological support were found to be useful in improving sleep quality after mTBI in two studies, including one large randomized controlled trial [37,38]. Light therapy also seems promising for improving sleep quality after mTBI [39,40].

### 4.3. Treatment of Somatic Complaints, Headaches, and Fatigue

Vestibular rehabilitation, cervical spine therapy, or techniques based on video games and virtual reality were found to improve balance disorders in four controlled studies, including two Grade A RCTs [41,42,43]. However, the persistence of a beneficial effect after the end of treatment seems questionable [41]. rTMS was found to be efficient for post-traumatic headaches after mTBI in three randomized sham-controlled studies [45,46,48] but one of them found effects which fell below clinical significance thresholds [47]. Thus, the level of evidence is low. Other interventions targeting post-traumatic headaches after mTBI, such as cognitive behavioral therapy [49] or manual therapy of the neck [50], reported negative results. Among three controlled studies targeting oculomotor disorders, an improvement was found regarding oculomotor rehabilitation in two studies by the same group [52,53], but hyperbaric oxygen treatment had no significant effect on eye movement disorders [54]. Finally, one Grade A RCT found that fatigue can be significantly alleviated by a graduated physical activity program [55].

### 4.4. Early Interventions

Eighteen controlled studies assessing the effect of an intervention within the first 3 months after the injury to prevent the occurrence or the persistence of PCS were selected; however, these studies provided mixed findings. Eight studies reported positive results with early interventions based on various combinations of providing written and oral information, reassurance, psycho-education, counseling (in person or by telephone or texting), and/or CBT [22,56,57,58,59,60,61,62]. In contrast, nine other early intervention studies failed to determine a beneficial effect. Some of these studies used multidisciplinary rehabilitation [64,65,68], and others relied on only one single early consultation [66,67,69] or on bed rest [63]. However, two well-designed RCTs based on multidimensional psycho-education, counseling, or CBT reported negative results [70,71]. It is thus difficult to draw any firm conclusions from these findings, although quite encouraging results were reported to suggest some beneficial preventive effects of a combination of information, psycho-education, reassurance, counseling, and CBT at the very early stage after mTBI.

In summary, the results of this review suggest that different rehabilitation programs, particularly cognitive training, psycho-education, telephone counseling, but also graded physical activity, could be efficient in decreasing persistent PCS of adult patients after mTBI. These results, although heterogeneous, support the use of a range of treatments for persistent PCS after mTBI and thus could provide guidance for healthcare professionals in the management of these patients and steer future studies. The present findings should encourage the development of evidence-based guidelines and information for patients, caregivers, and health professionals to improve global outcomes.

More precisely, it appeared that specific treatments could be useful to target different specific symptoms. In patients with predominant cognitive and global complaints, with or without associated PTSD, a combination of cognitive training and psycho-education could be useful. Anxiety and depression post-mTBI may be reduced using CBT. Sleep disorders may be improved by CBT or blue light therapy. Balance disorders could be at least temporarily improved by vestibular rehabilitation. Post-traumatic headaches could be reduced with rTMS. Fatigue can be alleviated by a graduated physical activity program. Finally, at-risk patients seen at the early stage (<3 months) could benefit from a program including psycho-education, reassurance, counseling, and/or CBT. In opposition, hyperbaric oxygen therapy, pharmaceutical drugs, and rTMS (for symptoms other than headaches) should not be recommended, given the available evidence.

The main strength of this review, based on 55 controlled studies (including 35 rank A), is the fact that it is, to the best of our knowledge, the first systematic literature review conducted according to the PRISMA guidelines on a large spectrum of PCS after mTBI, performed on several databases (Medline, Embase, and Cochrane), including research published up until August 2021. Other reviews have been published previously on specific aspects of PCS. For example, recent reviews [75,76,77] focused on the effect of physical exercise in patients with persistent PCS and found that exercise significantly reduced the severity of PCS, the percentage of patients with PCS, and days off work, as compared to controls. Other recent reviews addressed issues such as interventions in sport-related concussion [17,78], reporting evidence in support of cervical rehabilitation, vestibulo-ocular rehabilitation, aerobic exercise, or rTMS [51,79], suggesting promising preliminary results for the treatment of post-concussive depression and headaches. As previously mentioned, a systematic review with meta-analysis provided only very low to low levels of evidence to support commonly applied non-pharmacological interventions for persistent PCS [16]. The objective of the present study, in comparison with previous reviews, was to present a broad overview of the different possible interventions on the different facets of PCS rather than focusing on one single symptom or patient population.

However, this study has some limitations. First, there is probably a publication bias, as studies with negative results are often unpublished. We tried to minimize this bias by extending our search to databases other than Medline. In addition, several selected studies had a lower level of evidence, mainly because of the low number of participants included. Only seven studies had a low risk of bias with contrasting results, thus limiting the overall level of evidence, with the exception of blue light therapy, for which two low-risk of bias studies reported positive results. Furthermore, the studies were extremely heterogeneous in terms of rehabilitation type, the primary endpoint, outcome measures, time after trauma, and the number of patients; hence, meta-analysis was impossible, and this heterogeneity limits the potential conclusions. We did not find enough studies evaluating the impact of rehabilitation at a very long term (several years) after the mTBi; thus, we were unable to determine the maximum period of time during which rehabilitation could have a positive impact on persistent PCS. A final limitation is related to the wide variation in population samples and/or injury mechanisms included in the different studies in the present review (such as civilians, veterans, athletes, etc.). It was unfortunately not possible to untangle the effects of intervention in these different subpopulations, but this could be an important issue to consider in future research.

## 5. Conclusions

Despite these limitations, the results of the present review should encourage clinicians to propose tailored treatment to patients with persistent PCS, according to the type and severity of symptoms. For example, cognitive training and psycho-education could be recommended for patients with cognitive complaints and persistent PCS, cognitive behavioral therapy and light therapy for sleep disorders, rTMS for post-traumatic headaches after mTBI, a graduated physical activity program for persisting fatigue, and early counseling, reassurance, psycho-education, and/or CBT could be recommended for at-risk patients at an early stage (<3 months). Further research should be encouraged to assess such programs in larger groups of patients using a randomized controlled design.

## Figures and Tables

**Figure 1 jcm-11-06224-f001:**
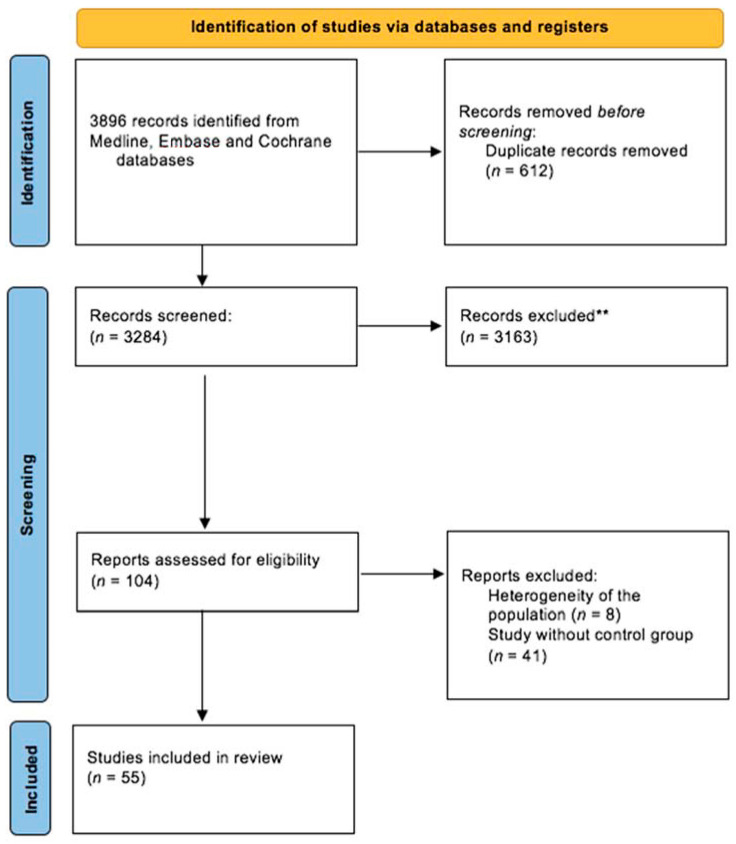
PRISMA Flow chart. ** main reasons for exclusion: studies conducted with children or adolescents, studies including a majority of participants with moderate to severe mTBI, and systematic reviews (screened for research of original articles).

## Data Availability

Not applicable.

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
