# Peer review of "A Systematic Review of Treatments of Post-Concussion Symptoms"

_jcm, 2022, doi:10.3390/jcm11206224_

Round 1

Reviewer 1 Report

General comments

The authors propose to review systematically the treatment of pos concussion symptoms describing in detail four separate domains of symptoms.  Through extensive citation they propose to grad the evidence and then present in sections dedicated to the symptom domain and through textual descriptions guide the reader through the available studies meeting the criteria decided upon for grad of evidence, etc. 

One of the first problems in any review like this is how should the authors , in the space allowed, describe each population form the studies cited.  It is absolutely crucial that these be listed and as written this could add a cumbersome amount of text.  Grouping by injury type would also destroy the flow established for the symptom domains making this a moderate to major weakness where the paper in general is very valuable for listing in textual form the wide array of treatments they review. 

Nevertheless, the paper is readable and well written.  The discussion is a restatement of much of the paper and could be either shortened or described in more general terms referring to text in the body of the paper.  The reader could, in fact, skip to the discussion and get most of what the authors are proposing so the editors should discuss with the authors whether this is necessary or recommended. 

Specific comments

Introduction: 

No comments or obvious weakness.  The format for the last paragraph numbering and bulleting is not effective visually and could be put into text without losing the guidance for the rest of the paper

Materials and Methods.  Well done and the figure is helpful.  Determining whether the search strategies and choices of terms is beyond the scope of this critique. 

P5.l 168 ff:  as above the choice of terms for each domain could be individually explored but this would entail deconstructing the entire paper.  It is VERY CURIOUS knowing the state of the literature and science of mTBI that migraine was not included as a search term. 

Results

Sections 3.x:  A major concern for this reader is that in reporting the outcomes of interest to the paper, it is difficult to find the populations of study and for this kind of paper, the reader would be specifically interested in the population but given the breadth of the studies it might not be practical.  OF IMPORTANT NOTE: I COULD NOT FIND TABLE 1

3.1.4  The authors conclude this section with a recommendation which is appreciated a to their best judgement of reading. 

3.2 p8.l332 ff.  the use of the term “recent TBI” here and in other places throughout the paper is too imprecise for a systematic review and should be corrected where used by including the time from injury even though stated in the supplemental material. 

3.3.1 p8/l343: “a few”{ could be replaced by the exact number in the supplemental materials referred to.

3.3.2:  those studies reviewed were well done though the grades of evidence could have been included with each study.  in addition, and just be3cuase of my focus, when I search PubMed using the search term “post traumatic headache” and “treatment” and systematic review filters 43 results come up.  I am not saying that the search terms were used differently or that the authors did not use additional modifiers, but it is curious and this reader wonders if the review has suffered because of choices made. 

https://pubmed.ncbi.nlm.nih.gov/?term=post+traumatic+headache+and+treatment&filter=pubt.systematicreview

3.5:  the figures accompanying the second on bias are interesting and show the information well that they highlighted in the introduction and methods, but it seems that this could have been a separate paper, and a short section without the figures would suffice. 

Discussion

As above the discussion basically retells the paper in more detail than needed.  The section on limitations I good to mention the populations, etc. and mechanics of injury and the statement “this could be an important issue” is an understatement at best.  Populations may in fact predispose to certain outcomes and probably needs to be accounted for.  This is major weakness that may be unavoidable in such a review.  The authors may need to consider a way to square this since the goal of the paper is to guide as well as inform.  

Author Response

General comments

Dear Reviewer1,

Thank you for considering our manuscript for publication in Journal of Clinical Medicine and for your helpful feedback. In the following sections, we have provided a detailed and point-by-point response to your comments and concerns (in blue). Corresponding changes are highlighted in yellow in the manuscript. We hope that the modifications done will satisfy you (please see the attachment file named Response to Reviewer 1).

The authors propose to review systematically the treatment of pos concussion symptoms describing in detail four separate domains of symptoms.  Through extensive citation they propose to grad the evidence and then present in sections dedicated to the symptom domain and through textual descriptions guide the reader through the available studies meeting the criteria decided upon for grad of evidence, etc. 

One of the first problems in any review like this is how should the authors, in the space allowed, describe each population form the studies cited.  It is absolutely crucial that these be listed and as written this could add a cumbersome amount of text.  Grouping by injury type would also destroy the flow established for the symptom domains making this a moderate to major weakness where the paper in general is very valuable for listing in textual form the wide array of treatments they review. 

We thank the Reviewer 1 for their positive comment about our work. We followed their suggestions and added in Table 1 a more precise description of the population included in each study included in terms of age, gender and main mechanisms of injury. Table 1 is now integrated at the end of the main manuscript.

Nevertheless, the paper is readable and well written.  The discussion is a restatement of much of the paper and could be either shortened or described in more general terms referring to text in the body of the paper.  The reader could, in fact, skip to the discussion and get most of what the authors are proposing so the editors should discuss with the authors whether this is necessary or recommended. 

We thank the Reviewer 1 for their comment and have now shortened the Discussion part and detailed the recommendations for Clinical practice that can be made based in the results of this review.

Specific comments

Introduction: 

No comments or obvious weakness.  The format for the last paragraph numbering and bulleting is not effective visually and could be put into text without losing the guidance for the rest of the paper

We deleted the numbering and bulleting of the last paragraph of the Introduction section.

Materials and Methods.  Well done and the figure is helpful.  Determining whether the search strategies and choices of terms is beyond the scope of this critique. 

Thank you for your positive comments.

P5.l 168 ff:  as above the choice of terms for each domain could be individually explored but this would entail deconstructing the entire paper.  It is VERY CURIOUS knowing the state of the literature and science of mTBI that migraine was not included as a search term. 

We thank the Reviewer 1 for this remark. We did a new research with migraine as a search term with the same other research protocol, and found for the same period (until August 2021) a new article (Esterov et al., 2021) which assessed the effect of osteopathic manipulative treatment on headache after mTBI. We added this new article to the Table 1, to the Results part and to the Discussion section.

Results

Sections 3.x:  A major concern for this reader is that in reporting the outcomes of interest to the paper, it is difficult to find the populations of study and for this kind of paper, the reader would be specifically interested in the population but given the breadth of the studies it might not be practical.  OF IMPORTANT NOTE: I COULD NOT FIND TABLE 1

We followed the recommendations of the Reviewer 1 and added in Table 1 a more precise description of the population included in each study included in terms of age, gender and mechanism of injury. The delay between the injury and the inclusion was already present in Table 1, as well as the type of study, number of participants included, main aim, main results, grade of recommendations. We are sorry that the Reviewer did not find Table 1 that we first sent separately and we have now included it in the main document.

3.1.4  The authors conclude this section with a recommendation which is appreciated a to their best judgement of reading. 

3.2 p8.l332 ff.  the use of the term “recent TBI” here and in other places throughout the paper is too imprecise for a systematic review and should be corrected where used by including the time from injury even though stated in the supplemental material. 

We have now replaced the term « recent TBI » by the exact delay contained in the articles quoted (less than 18 months in this case).

3.3.1 p8/l343: “a few”{ could be replaced by the exact number in the supplemental materials referred to.

We replaced « a few » by the exact number of articles included in Appendix A concerning dizziness and balance, i.e. 9.

3.3.2:  those studies reviewed were well done though the grades of evidence could have been included with each study.  in addition, and just be3cuase of my focus, when I search PubMed using the search term “post traumatic headache” and “treatment” and systematic review filters 43 results come up.  I am not saying that the search terms were used differently or that the authors did not use additional modifiers, but it is curious and this reader wonders if the review has suffered because of choices made. 

https://pubmed.ncbi.nlm.nih.gov/?term=post+traumatic+headache+and+treatment&filter=pubt.systematicreview

We checked the systematic reviews found by the Reviewer 1 by doing this research and did not find any new RCT to include to our results, most of the studies included in these reviews being uncontrolled studies, case series or case reports (already included in Appendix A).

3.5:  the figures accompanying the second on bias are interesting and show the information well that they highlighted in the introduction and methods, but it seems that this could have been a separate paper, and a short section without the figures would suffice. 

We now placed the figures concerning the risk of bias in Supplementary Material (Appendix B).

Discussion

As above the discussion basically retells the paper in more detail than needed.  The section on limitations I good to mention the populations, etc. and mechanics of injury and the statement “this could be an important issue” is an understatement at best.  Populations may in fact predispose to certain outcomes and probably needs to be accounted for.  This is major weakness that may be unavoidable in such a review.  The authors may need to consider a way to square this since the goal of the paper is to guide as well as inform.  

We thank the Reviewer 1 for their comment and have now shortened the discussion part and detailed the recommendations for Clinical practice that can be made based in the results of this review. We also included in the Discussion section a paragraph concerning the heterogeneity of the studies included in terms of population’s age, gender, type of injury and delay.

Reviewer 2 Report

This paper has a number of strengths. It covers both pharmacological and non–pharmacological interventions targeted to cognitive, physical and psychological symptoms of PCS after a mild traumatic brain injury. The paper appears to have faithfully followed PRISMA guidelines for conducting a systematic literature review. A quality assessment was conducted based on the Incog grading system. 54 studies were selected for review that had either grade A or B levels of evidence. The risk for bias in the selected studies was assessed using the Cochran Rob2 tool revised version.

Another strength of this paper is that 41 articles that were not included in the review because their quality grade was below B were presented in appendix a.

This paper, however, has major flaws. The most important flaw is the lack of transparency about the basis for the interpretation of the results of the papers included in the systematic review. The summaries of the papers included in the review are literally abstracts of abstracts. If the reader wanted information about the studies included in the review they would have to go to the original papers. This systematic review would make a much greater contribution to the field if key details of the papers, including results, were summarized in tables. Importantly, this would permit the reader to visualize the results of the papers included in the review and permit them to critically evaluate the conclusions and import of the results of studies included in the systematic review. Following the current outline of the text in the systematic review I would suggest the authors prepare separate tables for the three types of symptoms targeted (cognitive, physical, psychological). Each table should include at least the author, year of publication, the number of patients and controls, the target of the intervention, the primary outcome variable used to assess the effect of the intervention on the target, and the effect size of the treatment effect when that can be computed. (Note: I am not suggesting that the authors perform a metanalysis.) There may be some additional information that the authors decide to include in these tables. To make space for the tables results I suggested figures 2,3,4 be moved to the online supplement as appendix a.

Another significant shortcoming of this systematic review in its current form is that it does not compare the results/conclusions in this review to prior recent, more narrowly focused reviews.

As noted above, one of the potential strengths of this systematic review is the fact that it included both pharmacological and nonpharmacological interventions targeted to three broad classes of symptoms following concussion. However, in its current form the paper fails to capitalize on the broad scope of the papers reviewed. What is notable by its absence is any comparison, for example, of the differential effectiveness of treatments for the three symptoms. One would hope that what might emerge from the systematic review was a preliminary, cautious set of observations about what types of interventions seem to have the most promise for which class of symptoms.

Author Response

Reviewer 2

Dear Reviewer 2,

Thank you for considering our manuscript for publication in Journal of Clinical Medicine and for your helpful feedback. In the following sections, we have provided a detailed and point-by-point response to your comments and concerns (in blue). Corresponding changes are highlighted in yellow in the manuscript. We hope that the modifications done will satisfy you (please see the attachment named Response to Reviewer 2).

Suggestions for Authors

Overall: This manuscript presents the results of a systematic review which attempted to determine the effectiveness of post-concussion symptoms (PCS) treatments. The aim of the study is laudable given the conflicting information that is sometimes given to patients about the prevalence of and how to manage PCS. However, this paper has serious limitations which limit its impact. Please see below for specific comments. Additionally, the paper could benefit from a review by an editor – there were several small grammatical errors (e.g., mis-use of prepositions, mis-placement of apostrophes, extraneous words, extra commas).

We thank the Reviewer 2 for their positive comment about our work and now made the manuscript reviewed by a native English speaker, Anna Moraïtis.

Abstract:

  • Page 2, line 17: I think it should be “interventions FOR PCS.”

Modified

  • Page 2, line 21: I’m sure this will be covered in the methods, but it’s unclear how/why you went from over 3,200 abstracts to 121 full-text articles to 54 studies. The number of abstracts may not be necessary to cover here if you can’t explain it.

Deleted

Consider something such as “after review for x and y, we 54 studies were retained.”

  • Page 2, line 27: “remain up for debate?”

Modified

  • Page 2, lines 27-28: strongly suggest an alternative concluding sentence. Try finding something that better wraps up your entire study, not just another finding.

Modified, thank you: the sentence regarding the risk of bias has been removed earlier in the Abstract, and a more general conclusion has been added: « Despite its limitations, the results of the present review should encourage clinicians to propose a tailored treatment to patients, according to the type and severity of PCS, and could encourage further research. »

Introduction:

  • Page 2, first paragraph: consider restructuring the first sentences. E.g., first sentence: just the 42 million figure. Second sentence: definition of mTBI. Third sentence: prevalence of PCS. Fourth sentence: definition of PCS. It jumps back and forth currently.

Modified, thank you

  • Page 2, line 35: does WHO not require a hit to the head for their definition of mTBI?

Added

Carroll LJ, Cassidy JD, Holm L, Kraus J, Coronado VG, WHO Collaborating Centre Task Force on Mild Traumatic Brain Injury. Methodological issues and research recommendations for mild traumatic brain injury: the WHO Collaborating Centre Task Force on Mild Traumatic Brain Injury. J Rehabil Med 2004:113–25.

  • Page 3, lines 76-77: unclear why you are interested in studying bias here. Bias has not been mentioned in the introduction thus far.

Thank you for your remark. We have added a sentence in the Introduction part mentioning that the risk of bias of the studies included was not assessed in the latest systematic review performed in this field.

Materials and Methods:

  • Page 4, line 124: there is a double asterisk on “records excluded” but I don’t see a footnote for that. Really important to explain why the majority of records were excluded.

We added a footnote to detail the main reasons of exclusion of these records.

  • Page 5, line 153: please cite “incog grading system.”

Added

Bayley MT, Tate R, Douglas JM, Turkstra LS, Ponsford J, Stergiou-Kita M, Kua A, Bragge P; INCOG Expert Panel. INCOG guidelines for cognitive rehabilitation following traumatic brain injury: methods and overview. J Head Trauma Rehabil. 2014 Jul-Aug;29(4):290-306. doi: 10.1097/HTR.0000000000000070.

  • Page 5, line 170: unclear what “finally, 18 studies were included” refers to. Consider re-writing/re-framing this sentence. Same with line 173 and 176. You have “finally” three times in the paragraph. Also, none of these numbers appear in figure 1, although you are referring to it here. Consider lining up the text and figure better.

Modified, thank you.

  • Page 5, line 179: what are the 101 articles? This number does not appear in Figure 1. I see 121 articles.

 We corrected it in the manuscript.

Results:

  • Overall comment: the results present a LOT of information about each individual study. It’s difficult for the reader to digest and make sense of. The individual information would probably be more comprehensible in a table (such as what you propose in Appendix A) while the text could be reserved for a summary (e.g., 5 articles showed improvement in x after y therapy). I read the first results paragraph (which is probably too long and should be broken up) on page 6 and have a difficult time recollecting or making sense of what I read.

We are sorry that the Reviewer did not find Table 1 that we first sent separately and we have now included it in the main document. It contains the following elements for each article included: experimental design; age and gender of the participants; mechanisms of injury; time since injury; main objective of the study (including nature of the intervention and of control treatment when applicable); number of patients; main results. Quality assessment of the studies based on the Incog grading system.

Furthermore, we followed the recommendations of the Reviewer 1 and added in Table 1 a more precise description of the population included in each study included in terms of age, gender and mechanism of injury. The delay between the injury and the inclusion was already present in Table 1, as well as the type of study, number of participants included, main aim, main results, grade of recommendations.

Last, we summarized the Results parts in order to improve comprehension.

  • Page 7, line 269: what is rTMS? Please define.

Added (repetitive Transcranial Magnetic Stimulation)

  • Page 9, line 364: were these studies on post-mTBI headache or more generally post-traumatic headache? (other types of trauma). If it’s the latter, it’s unclear why these studies were included.

The studies included were only about post-mTBI headache. We made it clearer in the manuscript. Furthermore, as suggested by the Reviewer 1, we added the keyword « migraine » to our research.

  • Pages 10-11: the paragraphs on these pages are exceedingly long and complex. Suggest trying to break them up.

We now have summarized the Results section.

  • Page 11, lines 463, 467: the use of subjective words such as “unfortunately” is not standard practice in results sections.

Modified, thank you

  • Page 11, line 473: is 6-8 weeks post injury considered “early intervention?”

Yes, early interventions are defined by intervention within the 3 months after injury, as described in the first sentence of the paragraph.

  • Page 11, lines 490-495: it would be helpful to include information about these drugs. What are they meant to do? What were they originally designed to do?

This information is now included: « Cerebrolysin is a nootropic drug, which has been found useful to improve cognitive function in patients with Alzheimer ’s disease; and Atorvastatine has been found useful to improve cerebral plasticity. »

Discussion:

  • Overall comment: this is a comprehensive summary of the results and it could be used in the results section itself if a table was employed to provided the details of the studies (per my comment above). Otherwise, it is a bit too repetitive of the results section and doesn’t synthesize and interpret what you found. What are your recommendations to clinicians or people suffering from PCS? What additional research is needed?

Thank you for this remark. We have now added in the Discussion section a paragraph explaining more clearly how the results of the review could help clinicians to target treatments according to an individual patient’s symptoms:

“More precisely, it appeared that specific treatments could be useful to target different specific symptoms. In patients with predominant cognitive and global complaints, with or without associated PTSD, a combination of cognitive training and psychoeducation could be useful. Anxiety and depression post mTBI may be reduced using CBT. Sleep disorders may be improved by CBT or blue light therapy. Balance disorders could be at least temporarily improved by vestibular rehabilitation. Post-traumatic headaches could be reduced with rTMS. Fatigue can be alleviated by a graduated physical activity program. Finally, at-risk patients seen at the early (< 3 months) stage could benefit from a program including psycho-education, reassurance, counseling, and/or CBT. In opposition, hyberbaric oxygen therapy, pharmaceutical drugs, rTMS (for other symptoms than headache), should not be recommended given the available evidence.”

  • Page 15, line 621: what are the implications of the fact that most studies had a moderate or high risk of bias?

The following sentence has been added, in order to clarify this issue: “Only 7 studies had a low risk of bias, with contrasting results, thus limiting the overall level of evidence, with the exception of blue light therapy, for which two low-risk of bias studies reported positive results”.

  • Page 15, lines 634-636: your conclusion could be strengthened. I don’t really think the findings provide evidence that clinicians can propose a “tailored treatment” approach for patients with PCS. The findings were so heterogeneous that few conclusions can be drawn about what effective treatments should be proposed. Plus, what sort of “future research” should be done in particular?

Thank you for these suggestions. The conclusion has been modified as follows, to clarify these issues: “For example, cognitive training and psycho-education could be recommended for patients with cognitive complaints and persistent PCS, cognitive behavioural therapy and light therapy for sleep disorders, rTMS for post-traumatic headache, a graduated physical activity program for persisting fatigue, and early counseling, reassurance, psycho-education and/or CBT could be recommended for at-risk patients at an early (< 3 months) stage. Further research should be encouraged to assess such programs in larger group of patients using a randomised controlled design”.

Reviewer 3 Report

Overall: This manuscript presents the results of a systematic review which attempted to determine the effectiveness of post-concussion symptoms (PCS) treatments. The aim of the study is laudable given the conflicting information that is sometimes given to patients about the prevalence of and how to manage PCS. However, this paper has serious limitations which limit its impact. Please see below for specific comments. Additionally, the paper could benefit from a review by an editor – there were several small grammatical errors (e.g., mis-use of prepositions, mis-placement of apostrophes, extraneous words, extra commas).

Abstract:

·        Page 2, line 17: I think it should be “interventions FOR PCS.”

·        Page 2, line 21: I’m sure this will be covered in the methods, but it’s unclear how/why you went from over 3,200 abstracts to 121 full-text articles to 54 studies. The number of abstracts may not be necessary to cover here if you can’t explain it. Consider something such as “after review for x and y, we 54 studies were retained.”

·        Page 2, line 27: “remain up for debate?”

·        Page 2, lines 27-28: strongly suggest an alternative concluding sentence. Try finding something that better wraps up your entire study, not just another finding.

Introduction:

·        Page 2, first paragraph: consider restructuring the first sentences. E.g., first sentence: just the 42 million figure. Second sentence: definition of mTBI. Third sentence: prevalence of PCS. Fourth sentence: definition of PCS. It jumps back and forth currently.

·        Page 2, line 35: does WHO not require a hit to the head for their definition of mTBI?

·        Page 3, lines 76-77: unclear why you are interested in studying bias here. Bias has not been mentioned in the introduction thus far.

Materials and Methods:

·        Page 4, line 124: there is a double asterisk on “records excluded” but I don’t see a footnote for that. Really important to explain why the majority of records were excluded.

·        Page 5, line 153: please cite “incog grading system.”

·        Page 5, line 170: unclear what “finally, 18 studies were included” refers to. Consider re-writing/re-framing this sentence. Same with line 173 and 176. You have “finally” three times in the paragraph. Also, none of these numbers appear in figure 1, although you are referring to it here. Consider lining up the text and figure better.

·        Page 5, line 179: what are the 101 articles? This number does not appear in Figure 1. I see 121 articles.

Results:

·        Overall comment: the results present a LOT of information about each individual study. It’s difficult for the reader to digest and make sense of. The individual information would probably be more comprehensible in a table (such as what you propose in Appendix A) while the text could be reserved for a summary (e.g., 5 articles showed improvement in x after y therapy). I read the first results paragraph (which is probably too long and should be broken up) on page 6 and have a difficult time recollecting or making sense of what I read.

·        Page 7, line 269: what is rTMS? Please define.

·        Page 9, line 364: were these studies on post-mTBI headache or more generally post-traumatic headache? (other types of trauma). If it’s the latter, it’s unclear why these studies were included.

·        Pages 10-11: the paragraphs on these pages are exceedingly long and complex. Suggest trying to break them up.

·        Page 11, lines 463, 467: the use of subjective words such as “unfortunately” is not standard practice in results sections.

·        Page 11, line 473: is 6-8 weeks post injury considered “early intervention?”

·        Page 11, lines 490-495: it would be helpful to include information about these drugs. What are they meant to do? What were they originally designed to do?

Discussion:

·        Overall comment: this is a comprehensive summary of the results and it could be used in the results section itself if a table was employed to provided the details of the studies (per my comment above). Otherwise, it is a bit too repetitive of the results section and doesn’t synthesize and interpret what you found. What are your recommendations to clinicians or people suffering from PCS? What additional research is needed?

·        Page 15, line 621: what are the implications of the fact that most studies had a moderate or high risk of bias?

·        Page 15, lines 634-636: your conclusion could be strengthened. I don’t really think the findings provide evidence that clinicians can propose a “tailored treatment” approach for patients with PCS. The findings were so heterogeneous that few conclusions can be drawn about what effective treatments should be proposed. Plus, what sort of “future research” should be done in particular?

Author Response

Review 3

Dear Reviewer 3,

Thank you for considering our manuscript for publication in Journal of Clinical Medicine and for your helpful feedback. In the following sections, we have provided a detailed and point-by-point response to your comments and concerns (in blue). Corresponding changes are highlighted in yellow in the manuscript. We hope that the modifications done will satisfy you (please see the attachment named Response to Reviewer 3).

This paper has a number of strengths. It covers both pharmacological and non–pharmacological interventions targeted to cognitive, physical and psychological symptoms of PCS after a mild traumatic brain injury. The paper appears to have faithfully followed PRISMA guidelines for conducting a systematic literature review. A quality assessment was conducted based on the Incog grading system. 54 studies were selected for review that had either grade A or B levels of evidence. The risk for bias in the selected studies was assessed using the Cochran Rob2 tool revised version.

Another strength of this paper is that 41 articles that were not included in the review because their quality grade was below B were presented in appendix a.

We thank the Reviewer 3 for their positive comment about our work.

This paper, however, has major flaws. The most important flaw is the lack of transparency about the basis for the interpretation of the results of the papers included in the systematic review. The summaries of the papers included in the review are literally abstracts of abstracts. If the reader wanted information about the studies included in the review they would have to go to the original papers. This systematic review would make a much greater contribution to the field if key details of the papers, including results, were summarized in tables. Importantly, this would permit the reader to visualize the results of the papers included in the review and permit them to critically evaluate the conclusions and import of the results of studies included in the systematic review. Following the current outline of the text in the systematic review I would suggest the authors prepare separate tables for the three types of symptoms targeted (cognitive, physical, psychological). Each table should include at least the author, year of publication, the number of patients and controls, the target of the intervention, the primary outcome variable used to assess the effect of the intervention on the target, and the effect size of the treatment effect when that can be computed. (Note: I am not suggesting that the authors perform a metanalysis.) There may be some additional information that the authors decide to include in these tables. To make space for the tables results I suggested figures 2,3,4 be moved to the online supplement as appendix a.

We are sorry that the Reviewer did not find Table 1 that we first sent separately and we have now included it in the main document. It contains the following elements for each article included: experimental design; age and gender of the participants; mechanisms of injury; time since injury; main objective of the study (including nature of the intervention and of control treatment when applicable); number of patients; main results. Quality assessment of the studies based on the Incog grading system.

Furthermore, we followed the recommendations of the Reviewer 1 and added in Table 1 a more precise description of the population included in each study included in terms of age, gender and mechanism of injury.

Last, we summarized the Results parts in order to improve comprehension.

As suggested, we moved figures 2,3,4 to the online supplement as Appendix B.

Another significant shortcoming of this systematic review in its current form is that it does not compare the results/conclusions in this review to prior recent, more narrowly focused reviews.

The results of the present review have been now compared to other recent reviews, focusing on physical exercice, interventions in sport-related concussion, rTMS or non-pharmacological interventions (Rytter et al., 2021; Lal et al., 2018; Sullivan et al., 2018; Reid et al., 2021; Haider et al., 2021; Mollica et al., 2021): these are discussed in the Discussion section, in the paragraph starting with “The main strengths of this review…”.

As noted above, one of the potential strengths of this systematic review is the fact that it included both pharmacological and nonpharmacological interventions targeted to three broad classes of symptoms following concussion. However, in its current form the paper fails to capitalize on the broad scope of the papers reviewed. What is notable by its absence is any comparison, for example, of the differential effectiveness of treatments for the three symptoms. One would hope that what might emerge from the systematic review was a preliminary, cautious set of observations about what types of interventions seem to have the most promise for which class of symptoms.

Thank you for this suggestion. As indicated above (see responses to reviewer 2) we have now included a new paragraph in the Discussion and also an additional sentence in the conclusion to address the differential effectiveness of treatments on the different post-concussion symptoms.

Round 2

Reviewer 3 Report

I appreciate the author's attention to the reviewers' comments. The manuscript is much improved.